# Prescribing of antipsychotics among people with recorded personality disorder in primary care: a retrospective nationwide cohort study using The Health Improvement Network primary care database

Sarah Hardoon ![ORCID],[1] Joseph Hayes,[1] Essi Viding,[2] Eamon McCrory,[2] Kate Walters ![ORCID],[3] David Osborn[1]

¹Division of Psychiatry, UCL, London, UK
²Division of Psychology and Language Sciences, UCL, London, UK
³Department of Primary Care and Population Health, UCL, London, UK

**Correspondence to**
Prof David Osborn;
d.osborn@ucl.ac.uk

## ABSTRACT

**Objectives** To investigate the extent of antipsychotic prescribing to people with recorded personality disorder (PD) in UK primary care and factors associated with such prescribing.

**Design** Retrospective cohort study.

**Setting** General practices contributing to The Health Improvement Network UK-wide primary care database, 1 January 2000–31 December 2016.

**Participants** 46 210 people registered with participating general practices who had a record of PD in their general practice notes. 1358 (2.9%) people with missing deprivation information were excluded from regression analyses; no other missing data.

**Main outcome measures** Prescriptions for antipsychotics in general practice records and length of time in receipt of antipsychotic prescriptions.

**Results** Of 46 210 people with recorded PD, 15 562 (34%) were ever prescribed antipsychotics. Among the subgroup of 36 875 people with recorded PD, but no recorded severe mental illness (SMI), 9208 (25%) were prescribed antipsychotics; prescribing was lower in less deprived areas (adjusted rate ratio (aRR) comparing least to most deprived quintile: 0.56, 95% CI 0.48 to 0.66, p<0.001), was higher in females (aRR:1.25, 95% CI 1.16 to 1.34, p<0.001) and with a history of adverse childhood experiences (aRR:1.44, 95% CI 1.28 to 1.56, p<0.001). Median time prescribed antipsychotics was 605 days (IQR 197–1639 days). Prescribing frequency has increased over time.

**Conclusions** Contrary to current UK guidelines, antipsychotics are frequently and increasingly prescribed for extended periods to people with recorded PD, but with no history of SMI. An urgent review of clinical practice is warranted, including the effectiveness of such prescribing and the need to monitor for adverse effects, including metabolic complications.

## INTRODUCTION

The core features of personality disorder (PD) diagnoses are pervasive patterns of

behaviour and maladaptive traits leading to substantial personal distress, social dysfunction and disruption to others, often beginning in early adult life.[1] Prevalence of PD in the community is estimated to be 6%–10%, but estimates for PD vary globally and by clinical setting,[2] with limited research regarding the prevalence in primary care settings.[3] There is limited evidence on the effectiveness of antipsychotic pharmacotherapy in PD. Results from randomised controlled trials mainly focus on emotionally unstable PD (EUPD) populations, but the findings are limited by short-term follow-up, small sample sizes and variability in outcome measures.[4] In the UK, treatment guidelines (from the National Institute for Health and Care Excellence (NICE))

### Strengths and limitations of this study

► In the present report, we identified a very large UK-wide cohort of >46 000 people with a recorded personality disorder (PD) diagnosis, enabling, for the first time, a detailed UK-wide exploration of prescribing practices in PD over an 18-year period from 2000 to 2016.

exist for EUPD and dissocial PD. For both diagnoses it is recommended that 'drug treatment should not be used', apart from short-term sedating medication during crises in EUPD.[5] This is echoed by guidelines for EUPD from Australia's National Health and Medical Research Council.[6] Conversely, the American Psychiatric Association recommends targeted pharmacotherapy for PD: antidepressant medication for affective instability, mood stabilisers for impulsive aggression and antipsychotics for cognitive-perceptive disturbance.[7] Cochrane reviews find no evidence for the efficacy of antidepressant medication, but some evidence that mood stabilisers may reduce affective dysregulation and impulsive–aggression and that antipsychotic medication may improve cognitive–perceptual symptoms and affective dysregulation in PD.[8] Despite this mixed evidence, and counter to the NICE guidelines, there is some limited data suggesting that psychotropic prescribing for PD is occurring in the UK mental health service setting.[9] There is an urgent need to ascertain the wider extent and duration of psychotropic prescribing in PD. There is a particular concern regarding the prescribing of antipsychotics given their adverse event profile, including metabolic and cardiovascular risks.[10] Further, the characteristics of individuals prescribed psychotropic medications need to be determined to understand who may be most likely to receive these medications. For example, females are more likely than males to be diagnosed with borderline or EUPD and there is a high prevalence of adverse childhood experiences among those with EUPD.[11] Yet it is not known whether patients of particular gender, PD diagnosis or life history are more likely to receive antipsychotic medication. Time trends in antipsychotic prescribing for PD have not been examined and may be important, as UK mental health services and social care provision have been negatively affected by economic austerity.[12] This study aimed to determine how frequently antipsychotics are prescribed for people with a diagnosis of PD in UK primary care as well as factors associated with this prescribing. The three main aims of the study were: (1) To quantify the prevalence and duration of antipsychotic prescribing among patients with a record of PD, particularly those without any diagnostic record of severe mental illness (SMI); (2) To determine whether gender, age, area level deprivation, mental health comorbidities, category of PD diagnosis or a recorded history of adverse childhood experiences affect the likelihood of being prescribed antipsychotics and (3) To examine time trends in prescribing of antipsychotics.

## METHODS

### Study design

Retrospective nationwide cohort study.

### Data source

We used data from The Health Improvement Network (THIN) primary care database, which is a clinical database comprising, at time of analysis, computerised anonymised patient records retrieved from 744 member general practices across the UK, corresponding to approximately 6% of the UK population.[13] Patient records are longitudinal, including all information recorded by the general practitioner (GP) from when the patient registered with the member practice until the date the practice last contributed data to THIN or the patient left the practice. Records for a patient are linked via an anonymised patient identifier.

The database includes all diagnoses, symptoms and other health data (blood tests, health indicators) recorded by the GP in the patient's computerised records, as well as all prescriptions issued. Diagnoses, symptoms and other health information in THIN are coded using the Read code clinical classification system.[14] The database has been shown to be socially and demographically representative of the UK general practice population.[13]

### Study population

From the THIN database population, we identified the smaller study population (our PD cohort) comprising people registered with a participating THIN general practice for at least some part of the period 2000–2016, who had a record in their general practice notes at any time during or prior to this period of a diagnosis of PD (specific or non-specific), identified via the presence of a Read code indicative of a PD diagnosis. A list of relevant Read codes was developed using established search techniques[15] and with reference to International Classification of Diseases (ICD-10) chapters F60–F61. The diagnostic list was verified by a clinician (JH). The complete Read code list is provided in online supplemental table 1. All permanently registered patients were included from the date the general practice reached previously defined, established data quality thresholds: acceptable mortality rate date[16] and acceptable computer usage date.[17]

PD Read codes were categorised in line with current versions of the ICD-10 and the Diagnostic and Statistical Manual of Mental Disorders-5 as: paranoid, schizoid, dissocial, EUPD (including borderline), histrionic, anankastic, anxious/avoidant, dependent and other (chiefly comprising codes for non-specific PD, but also including diagnoses such as eccentric, Haltlose, immature, narcissistic, masochistic, passive-aggressive and mixed). Where an individual had more than one type of specific PD record, the most recent record was assigned as their diagnosis.

### Outcomes

The primary outcome was a prescription for antipsychotic medication in the individual's primary care records, any time after the initial PD diagnosis. We included all first generation and second generation antipsychotics in British National Formulary chapters 4.2.1; 4.2.2 (listed in online supplemental table 2).

It is not possible to ascertain from THIN records whether a prescription that was issued was actually dispensed or used by the individual. Therefore, we also included an

outcome of at least two consecutive prescriptions being issued fewer than 84 days apart as a likely indication that the first prescription had been used (corresponds to 3×28 days, a common period for which longer-term medicines are issued in UK primary care).

We also determined the length in days of continuous antipsychotic treatment. A continuous treatment period was defined as repeat prescriptions for antipsychotics no more than 84 days apart. The length of a continuous treatment period was then calculated from the date of the first prescription in the period to date of last prescription +84 days.

## Follow-up period
PD cohort entry was defined as the latest date of: registration at the general practice, the established practice acceptable mortality rate date,[16] the established practice acceptable computer usage date,[17] date of first PD diagnosis record, or 1 January 2000. End of follow-up for an individual was the earliest of: date of death, date patient transferred out of practice, latest date practice contributed data to THIN, or 31 December 2016.

## Covariates
Antipsychotic prescribing was explored according to the following sociodemographic characteristics: gender, age and Townsend score for area-level social deprivation for the area (postcode) in which the individual lived (grouped into quintiles).[18] The Townsend score incorporates four variables: percentage of adults unemployed, percentages non-car ownership, percentage non-home ownership and percentage overcrowded households.

We extracted data regarding comorbid mental health conditions defined as: (SMI, including schizophrenia, bipolar disorder and other non-organic psychotic illness),[19] depression, anxiety, phobia, hypochondria, obsessive–compulsive disorder, other neurosis, eating disorder, post-traumatic stress disorder. Each of these conditions were identified by the presence of a Read code at any time, using lists of Read codes developed through established techniques, with reference to relevant ICD chapters and verified by clinician (JH).

The presence of recorded adverse childhood experiences was identified from records with Read codes specifically relating to: contact with social services, child protection procedures, child on at risk register, child-at-risk case conference, out-of-home care or codes corresponding to items in the Centers for Disease Control and Prevention short adverse childhood experiences tool,[20] that is; direct reference to childhood maltreatment (physical or emotional abuse or neglect), or direct reference to household dysfunction (such as parental substance misuse). Codes included were either (1) codes recorded in childhood (when individual <17 years) or (2) codes recorded during adulthood indicative of a history of the above in childhood. The code list was based on previous work examining recording of childhood maltreatment in THIN.[21]

## Patient and public involvement
A group of service users was established in conjunction with the McPin Foundation, called the Pathfinder Data Science Patient and Public Involvement Group. Through regular meetings over the course of the study, members of the group were involved in the design, conduct and reporting plans for our research.

## Statistical analysis
We first determined the percentages of people: (1) receiving at least one antipsychotic prescription and (2) at least two antipsychotic prescriptions fewer than 84 days apart. We also determined percentages of people with a continuous treatment periods of at least (1) 6 months, (2) 1 year, (3) 3 years and (4) 5 years. Percentages were computed for the whole sample and then by: gender, age (in 10-year age bands), Townsend deprivation quintile, diagnostic category of PD, recorded mental health comorbidity and whether an individual had a recorded history of adverse childhood experiences. Analyses were stratified by whether patients ever had an additional recorded diagnosis of SMI. For the continuous treatment periods percentages calculations, the study sample was restricted to only individuals who had at least that length of follow-up in the database (6 months, 1 year, 3 years or 5 years).

We then used univariable and multivariable Poisson regression, with follow-up time for individuals as an offset, to estimate rate ratios for antipsychotic prescribing, to take account of differing follow-up times for individuals in the study. Rates were computed as total people receiving a prescription/ total follow-up time for people in each covariate category. Regressions were stratified by ever receiving a diagnosis of SMI. Multilevel Poisson models were used, with individuals nested in general practices, to account for potential clustering of individuals in general practices and clustering of prescribing practices within general practices.

Time trends in antipsychotic prescribing were determined as yearly prevalence estimates of antipsychotic prescriptions: namely the number within the sample with at least one antipsychotic prescription in that year/ number of the sample present in the database that year.

We calculated the median and IQR of length of longest continuous treatment period for all antipsychotics and for individual agents. The numbers of individuals receiving at least one prescription for individual agents were also computed.

We carried out two sensitivity analyses using different definitions of adverse childhood experiences. In the first sensitivity analysis, we restricted the definition of adverse childhood experiences to a record (current or retrospective) of contact with social services before age 17 years. In the second sensitivity analysis, we only included people with primary care data available from childhood (from at least age 11 years), to enable capture of adverse childhood experiences as recorded during childhood, rather than historically.

 

Stata (V.14) was used for all analyses.

## RESULTS

The Personality Disorder cohort comprised 46 210 people who had a record of PD in their general practice notes. Median duration in the study was 3.2 years (IQR 1.1–8.0 years). Just over half the PD cohort were female (23 528, 51%), and the average age at study entry was 43 years (table 1). The PD cohort had greater levels of deprivation relative to the general population, with 11 928 (26%) of people living in areas within the most deprived quintile, and 5523 (12%) in the least deprived quintile. The most commonly recorded diagnostic category of PD was EUPD recorded among 12 300 people (27% of the PD cohort), followed by anankastic (3648, 8%). However, 20 147 (44%) of the PD cohort had only non-specific PD records including primarily 14 321 (31%) with Read code E21.00 'PD' and 4676 (10%) with Read code E21z.00 'PD NOS (Not Otherwise Specified)' (online supplemental table 1). A record of ever being diagnosed with an SMI was present in 9335 people (20% of the PD cohort), while 34 522 (75%) and 23 595 (51%) had records of depression and anxiety respectively. A history of adverse childhood experiences was recorded for 4742 people (10%) in the PD cohort.

### Prevalence of antipsychotic prescribing

One-third of the PD cohort of people with recorded diagnosis of PD were prescribed antipsychotics at least once after their first PD record (15 562, 34%, table 2). The majority of those prescribed antipsychotics (9208, 59%) in the PD cohort had no record of SMI to explain the prescribing. Indeed, restricting the PD cohort to 36 875 individuals with recorded PD but no record of SMI, one quarter (9208, 25%) had been prescribed antipsychotics

Percentages of people in the PD cohort prescribed antipsychotics by gender, age, deprivation, mental health comorbidity, type of PD and recorded history of adverse childhood experiences, are shown in table 2. Among those with recorded PD but without recorded SMI, prescribing of antipsychotics was higher among females than males (5294 out of 18 882, 28% vs 3914 out of 17 993, 22%) and highest among those aged 20–34 years and over 75 years, compared with other age groups. Prescribing increased with deprivation, from 19% (894 out of 4630) in the least deprived quintile to 29% (2698 out of 9235) in the most deprived quintile. There was some variation in prescribing according to diagnostic category of PD, with prescribing highest among those with record of EUPD (3723 out of 9709, 38%), followed by paranoid PD (333 out of 1007, 33%). Those who had a recorded history of adverse childhood experiences in their GP record were more likely to be prescribed antipsychotics than those who did not (1184 out of 3796, 31% vs 8024 out of 33 079, 24%).

Adjusted and unadjusted rate ratios (aRR) comparing rates of prescribing of antipsychotics for different levels of the sociodemographic characteristics, are shown in

table 3. The differences in prescribing of antipsychotics by gender, age, and deprivation, type of PD and record of adverse childhood experience all remained significant after adjustment in the group with no additional record of SMI. The aRR comparing women to men was 1.25 (95% CI 1.16 to 1.34, p<0.001). The aRR comparing those over 75 years to those aged 35–44 years was 2.02 (95% CI 1.75 to 2.34, p<0.001). The aRR comparing the least to most deprived areas was 0.56 (95% CI 0.48 to 0.66, p<0.001). Rates for all diagnostic categories of PD were statistically significantly lower than for EUPD with aRRs relative to EUPD of between 0.19 (95% CI 0.15 to 0.23, p<0.001) for anankastic PD and 0.63 (95% CI 0.53 to 0.74, p<0.001) for paranoid PD. The aRR comparing history to no history of adverse childhood experiences was 1.41 (95% CI 1.28 to 1.56, p<0.001).

### Time trends

Figure 1 presents for each calendar year, the percentage of people in the PD cohort prescribed at least one antipsychotic in that year. The percentage of people receiving prescriptions increased over time: Of 6465 individuals who were present in the PD cohort in 2000, 1129 (17%) received antipsychotics in that year. Of 10 399 individuals present in the PD cohort in 2016, 2755 (26%) received antipsychotics (figure 1). Among those without a record of SMI, prescribing increased from 9% (484 out of 5268) in 2000 to 19% (1567 out of 8451) in 2016.

### Duration of antipsychotic treatment in PD

The median duration of continuous prescriptions of antipsychotics was 605 days (IQR 197–1639 days). Among those with no recorded SMI, the median duration of continuous treatment was 439 days (IQR 150–1218 days). For each of these results, we only assessed the longest period of treatment for each individual.

Among the total PD cohort of 46 210 people, 13 898 (30%) received at least two consecutive antipsychotic prescriptions within 84 days, and substantial numbers were prescribed antipsychotics over at least 6 months (11 906 out of 40 850 present for at least 6 months, 29%), 1 year (9588 out of 35 794, 27%), 3 years (5492 out of 24 081, 23%) and 5 years (3529 out of 17 588, 20%) (online supplemental table 3). Among the subgroup of individuals without recorded SMI, 22% (7925 out of 36 875) received at least two consecutive prescriptions within 84 days, and many were prescribed antipsychotics for more than 6 months (6548 out of 32 569, 20%), 1 year (4997 out of 28 576, 18%), 3 years (2538 out of 19 340, 13%) and 5 years (1485 out of 14 203, 11%).

### Individual antipsychotics agents

The most commonly prescribed antipsychotics in the PD cohort were Quetiapine (prescribed to 37%; 5819 of 15 562 individuals in receipt of antipsychotics), olanzapine (prescribed to 27%; 4269) and risperidone (22%; 3399) (online supplemental table 2). Similarly, among those without any record of SMI the three most commonly

**Table 1** Baseline characteristics of study cohort

| | All (n=46 210) | Recorded SMI (n=9335) | No recorded SMI (n=36 875) |
|---|---|---|---|
| | **Mean (SD)** | **Mean (SD)** | **Mean (SD)** |
| Age at study entry*, years | 42.65 (16.39) | 43.37 (15.24) | 42.47 (16.66) |
| | **N (%)** | **N (%)** | **N (%)** |
| **Gender** | | | |
| Men | 22 682 (49.1) | 4689 (50.2) | 17 993 (48.8) |
| Women | 23 528 (50.9) | 4646 (49.8) | 18 882 (51.2) |
| **Age, years** | | | |
| 0–15 | 384 (0.8) | 6 (0.1) | 378 (1.0) |
| 15–19 | 1729 (3.7) | 179 (1.9) | 1550 (4.2) |
| 20–24 | 4482 (9.7) | 707 (7.6) | 3775 (10.2) |
| 25–29 | 5089 (11.0) | 1004 (10.8) | 4085 (11.1) |
| 30–34 | 5312 (11.5) | 1178 (12.6) | 4134 (11.2) |
| 35–39 | 5541 (12.0) | 1291 (13.8) | 4250 (11.5) |
| 40–44 | 5251 (11.4) | 1214 (13.0) | 4037 (10.9) |
| 45–49 | 4379 (9.5) | 1002 (10.7) | 3377 (9.2) |
| 50–54 | 3802 (8.2) | 783 (8.4) | 3019 (8.2) |
| 55–59 | 3053 (6.6) | 596 (6.4) | 2457 (6.7) |
| 60–64 | 2335 (5.1) | 469 (5.0) | 1866 (5.1) |
| 65–69 | 1657 (3.6) | 311 (3.3) | 1346 (3.7) |
| 70–74 | 1152 (2.5) | 214 (2.3) | 938 (2.5) |
| 75–100 | 2044 (4.4) | 381 (4.1) | 1663 (4.5) |
| **Townsend deprivation score** | | | |
| 1 (least deprived) | 5523 (12.0) | 893 (9.6) | 4630 (12.6) |
| 2 | 6700 (14.5) | 1144 (12.3) | 5556 (15.1) |
| 3 | 9047 (19.6) | 1762 (18.9) | 7285 (19.8) |
| 4 | 11 654 (25.2) | 2563 (27.5) | 9091 (24.7) |
| 5 (most deprived) | 11 928 (25.8) | 2693 (28.8) | 9235 (25.0) |
| Missing | *1358 (2.9)* | *280 (3)* | *1078 (2.9)* |
| **ICD-10 diagnostic category of PD** | | | |
| Paranoid | 1988 (4.3) | 981 (10.5) | 1007 (2.7) |
| Schizoid | 1346 (2.9) | 565 (6.1) | 781 (2.1) |
| Dissocial | 2407 (5.2) | 536 (5.7) | 1871 (5.1) |
| Emotionally unstable | 12 300 (26.6) | 2591 (27.8) | 9709 (26.3) |
| Histrionic | 1168 (2.5) | 178 (1.9) | 990 (2.7) |
| Anankastic | 3648 (7.9) | 347 (3.7) | 3301 (9.0) |
| Anxious | 501 (1.1) | 62 (0.7) | 439 (1.2) |
| Dependent | 2705 (5.9) | 411 (4.4) | 2294 (6.2) |
| Other* | 20 147 (43.6) | 3664 (39.3) | 16 483 (44.7) |
| **Recorded psychiatric comorbidity** | | | |
| Any SMI | 9335 (20.2) | 9335 (100) | 0 (0) |
| Schizophrenia | *4105 (8.9)* | *4105 (44.0)* | *0 (0)* |
| Bipolar disorder | *2987 (6.5)* | *2987 (32.0)* | *0 (0)* |
| Other psychotic illness | *5071 (11.0)* | *5071 (54.3)* | *0 (0)* |
| Post-traumatic stress disorder | 1468 (3.2) | 306 (3.3) | 1162 (3.2) |
| Eating disorder | 2852 (6.2) | 613 (6.6) | 2239 (6.1) |

**Table 1** Continued

| | All (n=46 210) | Recorded SMI (n=9335) | No recorded SMI (n=36 875) |
|---|---|---|---|
| | Mean (SD) | Mean (SD) | Mean (SD) |
| Depression | 34 522 (74.7) | 7116 (76.2) | 27 406 (74.3) |
| Anxiety | 23 595 (51.1) | 5006 (53.6) | 18 589 (50.4) |
| Phobia | 3279 (7.10) | 666 (7.1) | 2613 (7.1) |
| Hypochondria | 416 (0.9) | 117 (1.3) | 299 (0.8) |
| Obsessive–compulsive disorder | 5333 (11.5) | 831 (8.9) | 4502 (12.2) |
| Other neurosis | 1353 (2.9) | 332 (3.6) | 1021 (2.8) |
| Recorded adverse childhood experience | | | |
| No record | 41 468 (89.7) | 8389 (89.9) | 33 079 (89.7) |
| Record | 4742 (10.3) | 946 (10.1) | 3796 (10.3) |

*Other PD includes: eccentric, Haltlose, immature, narcissistic, masochistic, passive-aggressive, mixed and non-specific.
ICD, International Classification of Diseases; PD, personality disorder; SMI, severe mental illness.

prescribed antipsychotics were: quetiapine (40%; 3657 of 9208 individuals in receipt of antipsychotics), olanzapine (21%; 1925) and risperidone (18%; 1660). Individuals may receive more than one type of antipsychotic agent so the percentages in online supplemental table 2 add up to more than 100%.

### Sensitivity analyses regarding the definition of adverse childhood experiences

We restricted the definition of adverse childhood experiences to a record occurring before age 17 years and a record of contact with social services. Of 1345 people who met this definition, 417 (31%) were prescribed antipsychotics. The rate ratios for all predictors of receiving antipsychotics were very similar to those when using the broader definition of adverse childhood experiences. People who met this definition had an aRR receiving antipsychotics of 1.45 (95% CI 1.22 to 1.71, p<0.001), compared with individuals without such a history in their health record.

In a second sensitivity analysis, we only included people with primary care data available from childhood (from at least age 11 years), to enable capture of adverse childhood experiences as recorded during childhood, rather than historically. There were 21 647 individuals with such data available (47% of our total PD cohort). Among these individuals, 1483 (7%) had records of social services contact from childhood, while 2805 (13%) had records in childhood referring to adverse childhood experiences (including but not limited to social services contact), and 3601 (17%) had records at any time referring to adverse childhood experiences (so including those with only retrospective records). The proportion of individuals within each of these groups prescribed antipsychotics were 34% (for childhood social service contact), 36% (record of trauma recorded in childhood) and 37% (record of childhood trauma recorded at any time). The aRR comparing those with and without social services contact in childhood, again among those without

recorded SMI was 1.36 (95% CI 1.15 to 1.60, p<0.001). In summary, there was a statistically significant higher frequency of antipsychotic prescribing for people with a record of adverse childhood experiences history, which remained despite more rigorous definitions of childhood trauma.

### DISCUSSION
### Principal findings

In this large UK-wide cohort of over 46 000 people with a record of PD diagnosis in their general practice notes, we found that more than one-third of people with PD were prescribed antipsychotics in UK primary care. Prescribing was common among people with PD even when there was no record of a diagnosis of SMI for which antipsychotics are usually indicated. These antipsychotic medications were prescribed for considerable lengths of time (frequently over a year) and prescribing has become more common over the period of our study to 2016. Among those with recorded PD but without recorded SMI, antipsychotic prescriptions were more common among women, individuals living in areas with greater deprivation, individuals with a record of EUPD or paranoid PD, and where the individual also had a history of adverse childhood experiences. The finding regarding adverse childhood experiences remained robust despite several analyses restricting the definition of adverse childhood experiences.

### Comparison with other studies

To our knowledge, this is the first study to investigate the pattern of antipsychotic prescribing among people with records of PD in UK primary care. Our findings suggest that while NICE guidance does not recommend long term prescribing of antipsychotics in treatment of PD, this practice is common in real life samples.[5] There is some evidence that particular symptoms occurring within PD may be treated by antipsychotics: impulsivity,

**Table 2** Proportions of individuals receiving at least one prescription for antipsychotics during follow-up

| | Total | Prescribed antipsychotics, n (%) | | Recorded SMI | Prescribed antipsychotics, n (%) | | No recorded SMI | Prescribed antipsychotics, n (%) | |
|---|---|---|---|---|---|---|---|---|---|
| All | 46 210 | 15 562 | (33.7) | 9335 | 6354 | (68.1) | 36 875 | 9208 | (25.0) |
| Gender | | | | | | | | | |
| Men | 22 682 | 6958 | (30.7) | 4689 | 3044 | (64.9) | 17 993 | 3914 | (21.8) |
| Women | 23 528 | 8604 | (36.6) | 4646 | 3310 | (71.2) | 18 882 | 5294 | (28.0) |
| Age, years | | | | | | | | | |
| 0–15 | 384 | 26 | (6.8) | 6 | 4 | (66.7) | 378 | 22 | (5.8) |
| 15–19 | 1729 | 556 | (32.2) | 179 | 127 | (70.9) | 1550 | 429 | (27.7) |
| 20–24 | 4482 | 1543 | (34.4) | 707 | 450 | (63.6) | 3775 | 1093 | (29.0) |
| 25–29 | 5089 | 1866 | (36.7) | 1004 | 665 | (66.2) | 4085 | 1201 | (29.4) |
| 30–34 | 5312 | 2009 | (37.8) | 1178 | 835 | (70.9) | 4134 | 1174 | (28.4) |
| 35–39 | 5541 | 2068 | (37.3) | 1291 | 905 | (70.1) | 4250 | 1163 | (27.4) |
| 40–44 | 5251 | 1924 | (36.6) | 1214 | 840 | (69.2) | 4037 | 1084 | (26.9) |
| 45–49 | 4379 | 1503 | (34.3) | 1002 | 718 | (71.7) | 3377 | 785 | (23.2) |
| 50–54 | 3802 | 1169 | (30.7) | 783 | 524 | (66.9) | 3019 | 645 | (21.4) |
| 55–59 | 3053 | 816 | (26.7) | 596 | 421 | (70.6) | 2457 | 395 | (16.1) |
| 60–64 | 2335 | 586 | (25.1) | 469 | 308 | (65.7) | 1866 | 278 | (14.9) |
| 65–69 | 1657 | 420 | (25.3) | 311 | 190 | (61.1) | 1346 | 230 | (17.1) |
| 70–74 | 1152 | 337 | (29.3) | 214 | 128 | (59.8) | 938 | 209 | (22.3) |
| 75–100 | 2044 | 739 | (36.2) | 381 | 239 | (62.7) | 1663 | 500 | (30.1) |
| Townsend deprivation score | | | | | | | | | |
| 1 (least deprived) | 5523 | 1486 | (26.9) | 893 | 592 | (66.3) | 4630 | 894 | (19.3) |
| 2 | 6700 | 1932 | (28.8) | 1144 | 772 | (67.5) | 5556 | 1160 | (20.9) |
| 3 | 9047 | 2956 | (32.7) | 1762 | 1204 | (68.3) | 7285 | 1752 | (24.0) |
| 4 | 11 654 | 4176 | (35.8) | 2563 | 1751 | (68.3) | 9091 | 2425 | (26.7) |
| 5 (most deprived) | 11 928 | 4559 | (38.2) | 2693 | 1861 | (69.1) | 9235 | 2698 | (29.2) |
| Missing | *1358* | *453* | *(33.4)* | *280* | *174* | *(62.1)* | *1078* | *279* | *(25.9)* |
| ICD-10 diagnostic category of PD | | | | | | | | | |
| Paranoid | 1988 | 1026 | (51.6) | 981 | 693 | (70.6) | 1007 | 333 | (33.1) |
| Schizoid | 1346 | 536 | (39.8) | 565 | 394 | (69.7) | 781 | 142 | (18.2) |
| Dissocial | 2407 | 763 | (31.7) | 536 | 337 | (62.9) | 1871 | 426 | (22.8) |
| Emotionally unstable | 12 300 | 5573 | (45.3) | 2591 | 1850 | (71.4) | 9709 | 3723 | (38.3) |
| Histrionic | 1168 | 285 | (24.4) | 178 | 116 | (65.2) | 990 | 169 | (17.1) |
| Anankastic | 3648 | 700 | (19.2) | 347 | 223 | (64.3) | 3301 | 477 | (14.5) |
| Anxious | 501 | 130 | (25.9) | 62 | 42 | (67.7) | 439 | 88 | (20.0) |
| Dependent | 2705 | 686 | (25.4) | 411 | 303 | (73.7) | 2294 | 383 | (16.7) |
| Other | 20 147 | 5863 | (29.1) | 3664 | 2396 | (65.4) | 16 483 | 3467 | (21.0) |
| Psychiatric comorbidity | | | | | | | | | |
| Any SMI | 9335 | 6354 | (68.1) | 9335 | 6354 | (68.1) | 0 | 0 | (.) |
| Schizophrenia | 4105 | 2981 | (72.6) | 4105 | 2981 | (72.6) | 0 | 0 | (.) |
| Bipolar disorder | 2987 | 2044 | (68.4) | 2987 | 2044 | (68.4) | 0 | 0 | (.) |
| Other psychotic illness | 5071 | 3583 | (70.7) | 5071 | 3583 | (70.7) | 0 | 0 | (.) |
| Post-traumatic stress disorder | 1468 | 731 | (49.8) | 306 | 223 | (72.9) | 1162 | 508 | (43.7) |
| Eating disorder | 2852 | 1233 | (43.2) | 613 | 448 | (73.1) | 2239 | 785 | (35.1) |
| Depression | 34 522 | 12 698 | (36.8) | 7116 | 4979 | (70.0) | 27 406 | 7719 | (28.2) |
| Anxiety | 23 595 | 9201 | (39.0) | 5006 | 3601 | (71.9) | 18 589 | 5600 | (30.1) |

Continued

**Table 2** Continued

| | Total | Prescribed antipsychotics, n (%) | | Recorded SMI | Prescribed antipsychotics, n (%) | | No recorded SMI | Prescribed antipsychotics, n (%) | |
|---|---|---|---|---|---|---|---|---|---|
| Phobia | 3279 | 1272 | (38.8) | 666 | 470 | (70.6) | 2613 | 802 | (30.7) |
| Hypochondria | 416 | 176 | (42.3) | 117 | 81 | (69.2) | 299 | 95 | (31.8) |
| Obsessive–compulsive disorder | 5333 | 1517 | (28.4) | 831 | 560 | (67.4) | 4502 | 957 | (21.3) |
| Other neurosis | 1353 | 466 | (34.4) | 332 | 245 | (73.8) | 1021 | 221 | (21.6) |
| No recorded psychiatric comorbidity | 5763 | 874 | (15.2) | 0 | 0 | (.) | 5763 | 874 | (15.2) |
| Recorded adverse childhood experiences | | | | | | | | | |
| No record | 41 468 | 13 798 | (33.3) | 8389 | 5774 | (68.8) | 33 079 | 8024 | (24.3) |
| Record | 4742 | 1764 | (37.2) | 946 | 580 | (61.3) | 3796 | 1184 | (31.2) |

ICD-10, International Classification of Diseases-10th revision; PD, personality disorder; SMI, severe mental illness.

aggression, poor interpersonal relationships, global functioning and cognitive-perceptual symptoms are improved in meta-analyses of trials in EUPD.[22 23] It is potentially understandable that clinicians would extrapolate these findings to the broader PD population in lieu of psychological treatments which may be harder to access for those with PD or may not be the patient's preference for treatment. However, in our study, these antipsychotics are clearly being prescribed for considerable lengths of time and these treatments may not be reviewed or monitored appropriately, including for metabolic side effects.

We are not aware of any previous studies which examine antipsychotic medication according to history of adverse childhood experiences in people with PD. However, previous studies in SMI found that higher adverse childhood experience questionnaire scores are associated with higher antipsychotic prescribing, irrespective of diagnosis.[24] Our findings are in line with this evidence from other diagnostic categories, but primary care data do not permit interrogation of the decision-making process underlying prescribing.

### Strengths and weaknesses

The strengths of our study include the large size of the PD cohort, enabling precise estimates and UK-wide coverage. While the THIN database pertains to approximately 6% of the UK population, previous studies have shown the THIN database to be socially and demographically representative of wider UK general practice,[13] thus, the results may be cautiously generalised beyond the THIN database population to the wider UK general practice population. The study also benefits from complete records on the outcome (antipsychotics prescribed in primary care, including the type of antipsychotic), as all prescriptions are automatically computerised. Antipsychotics administered in secondary care, including some depot injections and clozapine would not be captured; the numbers in receipt of antipsychotics could be even higher if medication received in secondary care was also considered. We were not able to extract information on alternative

treatment options, such as psychological therapy. All individuals with any record in their general practice notes of PD were included in the PD cohort. It is not possible to ascertain how the PD diagnoses were arrived at or to verify them. Therefore, there may be people in the PD cohort misclassified as having PD and some people with these problems will not be coded in the database. Similarly, only cases of SMI and adverse childhood experiences recorded using Read codes are captured. However, a previous study found prevalence of SMI diagnosis in THIN to be consistent with other epidemiological findings, suggesting that under recording is minimal.[19] It is likely that some adverse childhood experiences will not be captured in the THIN data, where either the individual has not told their GP or the GP has not recorded this history as a Read code. The impact would be to underestimate the rate ratio for antipsychotic prescribing comparing those with and without adverse childhood experiences. However, sensitivity analyses restricting to a tighter definition of childhood contact with social services and limiting the PD cohort to those with records before aged 18 years and before aged 11 years did not meaningfully influence our main findings. A further limitation is that the clusters of PD severity used in other international settings are not coded in UK primary care. Results were not broken down by type of antipsychotic (first or second generation). However, since second generation antipsychotics are far more commonly prescribed in the cohort (online supplemental table), we would expect the patterns in prescribing of all antipsychotics to largely reflect patterns for second generation antipsychotics

### Interpretation, implications, conclusions and future research

We found that a considerable proportion of people with a record of PD in primary care had been prescribed antipsychotics, even when there was no recorded alternative indication (SMI record). A number of individuals received these medications for an extended period of time (at least 5 years). This is potentially concerning as longer term prescribing of antipsychotics is counter

**Table 3** Rate ratios for receiving at least one antipsychotic prescription any time after first PD record

| Predictor | Recorded SMI | | | | No recorded SMI | | | |
|---|---|---|---|---|---|---|---|---|
| | Univariable models | | Multivariable models* | | Univariable models | | Multivariable models* | |
| | RR (95% CI) | P value | RR (95% CI) | P value | RR (95% CI) | P value | RR (95% CI) | P value |
| Gender | | | | | | | | |
| Men | 1 | | 1 | | 1 | | 1 | |
| Women | 1.29 (1.15 to 1.45) | <0.001 | 1.17 (1.04 to 1.32) | 0.01 | 1.47 (1.34 to 1.61) | <0.001 | 1.25 (1.16 to 1.34) | <0.001 |
| Age, years | | | | | | | | |
| 0–15 | 0.2 (0.05 to 0.72) | 0.01 | 0.18 (0.05 to 0.58) | 0.004 | 0.12 (0.08 to 0.20) | <0.001 | 0.14 (0.08 to 0.22) | <0.001 |
| 15–24 | 1.03 (0.85 to 1.25) | 0.8 | 0.89 (0.73 to 1.08) | 0.2 | 1.63 (1.45 to 1.83) | <0.001 | 1.14 (1.01 to 1.30) | 0.04 |
| 25–34 | 1.21 (1.03 to 1.43) | 0.02 | 1.13 (0.96 to 1.33) | 0.1 | 1.38 (1.28 to 1.50) | <0.001 | 1.17 (1.06 to 1.30) | 0.002 |
| 35–44 | 1 | | 1 | | 1 | | 1 | |
| 45–54 | 0.99 (0.84 to 1.17) | 0.9 | 1.06 (0.89 to 1.25) | 0.5 | 0.68 (0.62 to 0.73) | <0.001 | 0.78 (0.71 to 0.85) | <0.001 |
| 55–64 | 0.77 (0.63 to 0.95) | 0.01 | 0.86 (0.70 to 1.06) | 0.2 | 0.41 (0.36 to 0.46) | <0.001 | 0.53 (0.45 to 0.61) | <0.001 |
| 65–74 | 0.68 (0.52 to 0.89) | 0.004 | 0.75 (0.57 to 0.98) | 0.04 | 0.55 (0.47 to 0.65) | <0.001 | 0.74 (0.62 to 0.89) | 0.001 |
| 75–100 | 0.99 (0.76 to 1.29) | 0.9 | 1.1 (0.84 to 1.44) | 0.5 | 1.52 (1.31 to 1.75) | <0.001 | 2.02 (1.75 to 2.34) | <0.001 |
| | | <0.001† | | 0.003† | | <0.001† | | <0.001† |
| Townsend deprivation score | | | | | | | | |
| 1 (least deprived) | 0.75 (0.60 to 0.93) | 0.01 | 0.79 (0.63 to 0.98) | 0.03 | 0.46 (0.38 to 0.56) | <0.001 | 0.56 (0.48 to 0.66) | <0.001 |
| 2 | 0.83 (0.67 to 1.03) | 0.09 | 0.88 (0.71 to 1.08) | 0.2 | 0.53 (0.47 to 0.60) | <0.001 | 0.64 (0.57 to 0.71) | <0.001 |
| 3 | 0.85 (0.72 to 1.00) | 0.06 | 0.87 (0.74 to 1.03) | 0.1 | 0.7 (0.63 to 0.76) | <0.001 | 0.78 (0.71 to 0.84) | <0.001 |
| 4 | 0.89 (0.78 to 1.03) | 0.1 | 0.89 (0.78 to 1.02) | 0.1 | 0.87 (0.80 to 0.94) | 0.001 | 0.9 (0.83 to 0.97) | 0.007 |
| 5 (most deprived) | 1 | | 1 | | 1 | | 1 | |
| | | 0.08† | | 0.2† | | <0.001† | | <0.001† |
| ICD-10 diagnostic category of PD | | | | | | | | |
| Anankastic | 0.32 (0.23 to 0.44) | <0.001 | 0.35 (0.26 to 0.48) | <0.001 | 0.14 (0.12 to 0.16) | <0.001 | 0.19 (0.16 to 0.21) | <0.001 |
| Anxious | 0.44 (0.25 to 0.80) | 0.006 | 0.47 (0.26 to 0.85) | 0.01 | 0.23 (0.15 to 0.35) | <0.001 | 0.29 (0.20 to 0.41) | <0.001 |
| Emotionally unstable | 1 | | 1 | | 1 | | 1 | |
| Dependent | 0.57 (0.41 to 0.78) | 0.001 | 0.62 (0.44 to 0.86) | 0.004 | 0.17 (0.14 to 0.21) | <0.001 | 0.22 (0.17 to 0.27) | <0.001 |
| Dissocial | 0.6 (0.46 to 0.80) | <0.001 | 0.66 (0.49 to 0.88) | 0.005 | 0.34 (0.28 to 0.40) | <0.001 | 0.45 (0.39 to 0.52) | <0.001 |
| Histrionic | 0.35 (0.24 to 0.51) | <0.001 | 0.37 (0.25 to 0.54) | <0.001 | 0.15 (0.13 to 0.19) | <0.001 | 0.19 (0.15 to 0.23) | <0.001 |
| Other | 0.54 (0.47 to 0.62) | <0.001 | 0.57 (0.49 to 0.67) | <0.001 | 0.27 (0.25 to 0.30) | <0.001 | 0.33 (0.30 to 0.36) | <0.001 |
| Paranoid | 0.68 (0.55 to 0.83) | <0.001 | 0.72 (0.57 to 0.91) | 0.005 | 0.52 (0.44 to 0.61) | <0.001 | 0.63 (0.53 to 0.74) | <0.001 |
| Schizoid | 0.42 (0.32 to 0.53) | <0.001 | 0.46 (0.35 to 0.60) | <0.001 | 0.18 (0.14 to 0.23) | <0.001 | 0.24 (0.18 to 0.33) | <0.001 |

Continued

**Table 3** Continued

| Predictor | Recorded SMI | | | | No recorded SMI | | | |
| --- | --- | --- | --- | --- | --- | --- | --- | --- |
| | Univariable models | | Multivariable models* | | Univariable models | | Multivariable models* | |
| | RR (95% CI) | P value | RR (95% CI) | P value | RR (95% CI) | P value | RR (95% CI) | P value |
| Recorded adverse childhood experiences | | <0.001† | | <0.001† | | <0.001† | | <0.001† |
| No record | 1 | | 1 | | 1 | | 1 | |
| Record | 1.1 (0.92 to 1.31) | 0.3 | 0.93 (0.78 to 1.10) | 0.4 | 2.32 (2.03 to 2.65) | <0.001 | 1.41 (1.28 to 1.56) | <0.001 |

From Poisson regression, with generalised estimating equations to account for potential clustering of patients in general practices.
*Adjusted for all the other characteristics considered.
†P values in italics indicate significance of categorical variables as a whole, as opposed to the individual categories.
ICD-10, International Classification of Diseases-10th revision; PD, personality disorder.

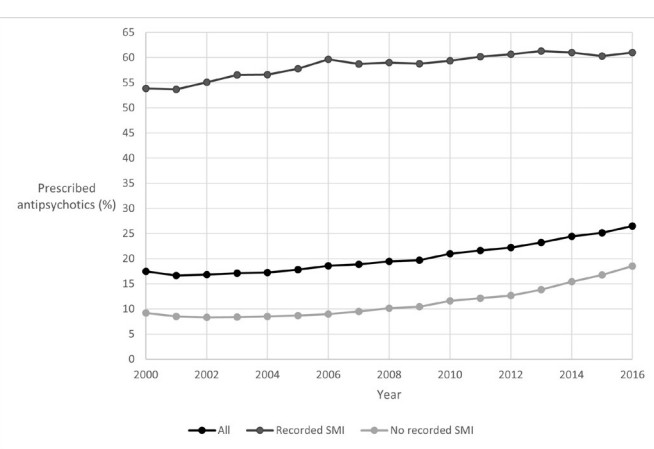

**Figure 1** Percentage of cohort prescribed antipsychotics in each calendar year*. *Percentage of cohort prescribed antipsychotics in each calendar year=100 × (number of people from cohort present in that calendar year with at least one antipsychotic prescription in that year)/(number of people from cohort present in that calendar year). SMI, severe mental illness.

to current UK NICE guidelines, which do not support routine use of antipsychotics to treat PD (in the absence of SMI), on account of the limited evidence regarding their effectiveness for PD.[5] The people in our PD cohort receiving antipsychotics will be at risk of experiencing adverse side effects from the drugs, possibly without any proven long term benefit of these agents for their mental health. They may develop adverse physical effects and yet may not receive appropriate physical health monitoring for weight gain and metabolic disturbance, because in the UK National Health Service physical health monitoring is currently only targeted at people with SMI. The lack of formal monitoring and protocols for prescribing antipsychotics in PD could also put people at risk of adverse effects of polypharmacy (eg, other psychotropic medications or medications for physical long-term conditions).

Possible explanations for the use of antipsychotics in PD could include lack of availability or long delays in access to other evidence based psychological therapies or because previous therapy/other treatments have not helped. We were unable to determine receipt of psychological therapies for emotional dysregulation and other problems as these therapies are usually provided in secondary care and not well recorded in primary care databases such as THIN. The use of antipsychotics could reflect patient or practitioner preference, due to belief in the effectiveness of antipsychotics for PD or indeed diagnostic uncertainty where mood instability or transient psychotic symptoms are present. It may be that the prescriptions issued in primary care are frequently provided as follow-on prescribing, after an initial prescription by a secondary care psychiatrist, without clear guidance on when to stop prescribing the antipsychotic drugs. The increase in antipsychotic prescriptions over time could reflect the impact of longer waiting lists for psychological interventions during a period of economic austerity, as well as general

trends in prescribing of antipsychotics that has been observed for other conditions, and increased reporting of adverse childhood experiences, which were associated with higher levels of prescribing.[25]

There were variations in antipsychotic prescribing levels by sociodemographic characteristics. Notably, prescribing was higher in the more deprived areas, which could reflect poorer access to psychological therapies or higher levels of distress. Prescribing was also highest among those aged over 75 years, who would be particularly susceptible to the adverse metabolic effects of antipsychotics. Studies have previously shown antipsychotics being prescribed for dementia,[26] despite also not recommended, which could explain some of the prescribing of antipsychotics in this age group.

The higher levels of prescribing in individuals with histories of adverse childhood experiences are not possible to explain fully with routine primary care data. People with records of adverse childhood experiences were more likely have a record of EUPD, compared with other PD types (almost half of all people with adverse childhood experiences, compared with one-quarter of those without), and were living in more deprived areas, both of these characteristics which were also associated with higher levels of antipsychotic prescriptions in our PD cohort. However, the increased rate of prescribing among people with adverse childhood experiences was observed even in multivariable models. This pattern is of particular concern as it is likely those with adverse childhood experiences might benefit more from appropriate psychological interventions as opposed to long-term medications.[27]

Further investigations are warranted to understand the reasons for longer term prescribing of antipsychotics in PD in primary care, particularly among those groups with higher prescribing levels. Future work should identify ways to address this and to promote recommended care pathways. Investigations of antipsychotic prescribing according to PD severity clusters is also warranted. Further research is needed to investigate the short and longer term effectiveness of antipsychotics in PD and the potential for adverse events in this group as well as the delivery of monitoring of such effects in primary care. This work would inform future guidelines regarding treatments for people with PD, to ultimately ensure people with PD receive the most appropriate, safe and effective treatment.

**Acknowledgements** The Pathfinder Data Science PPI Group, led by Dan Robotham, from the McPin Foundation for their input and feedback on the study. DO and JH are supported by the University College London Hospitals National Institute for Health Research (NIHR) Biomedical Research Centre and the NIHR North Thames Applied Research Collaboration.

**Contributors** SH: formulation of research questions, study design, extraction and analysis of data, interpretation of data, writing of first draft of paper, critical revision of paper for important intellectual content, approval of final draft. JH: formulation of research questions, study design, acquisition of data, interpretation of data, writing of first draft of paper, critical revision of paper for important intellectual content, approval of final draft. EV: formulation of research questions, study design, interpretation of data, critical revision of paper for important intellectual content, approval of final draft. EM: formulation of research questions, study design, interpretation of data, critical revision of paper for important intellectual content, approval of final draft. KRW: interpretation of data, critical revision of paper for important intellectual content, approval of final draft. DO: formulation of research questions, study design, interpretation of data, critical revision of paper for important intellectual content, approval of final draft. SH: Guarantor.

**Funding** This work was supported by the Medical Research Council (grant number MC_PC_17216); and the Wellcome Trust (JFH, grant number 211085/Z/18/Z).

**Disclaimer** This funder had no role in study design, data collection, data analysis, data interpretation, or writing of the report. The views expressed in this article are those of the authors and not necessarily those of the NHS, the NIHR, or the Department of Health and Social Care.

**Competing interests** None declared.

**Patient consent for publication** Not applicable.

**Ethics approval** Anonymised data were used throughout the study provided by the data provider to UCL. Studies using The Health Improvement Network (THIN) database have had initial ethical approval from the NHS South-East Multicentre Research Ethics Committee, subject to prior independent scientific review. The Scientific Review Committee (IQVIA) approved the study protocol (SRC Reference Number: 17THIN072) prior to its undertaking.

**Provenance and peer review** Not commissioned; externally peer reviewed.

**Data availability statement** No data are available. No data are available as no new data collected. Read code list for personality disorder in online supplemental material.

**ORCID iDs**
Sarah Hardoon http://orcid.org/0000-0002-3044-3713
Kate Walters http://orcid.org/0000-0003-2173-2430

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
