## [Reviewer comments · BMJ Open]

ARTICLE DETAILS

TITLE (PROVISIONAL)	Prescribing of antipsychotics among people with recorded Personality Disorder in primary care: a retrospective nationwide cohort study using The Health Improvement Network primary care database
AUTHORS	Hardoon, Sarah; Hayes, Joseph; Viding, Essi; McCrory, Eamon; Walters, Kate; Osborn, David

VERSION 1 – REVIEW

REVIEWER	Ponizovsky, Alexander M. Israeli Ministry of Health
REVIEW RETURNED	16-Aug-2021

GENERAL COMMENTS	Minor points 1. All abbreviations should be fully explained in the first mention in the text. 2. Abstract. Investigate should be read To investigate... 3. Introduction. Aims 2 is duplicated (see aim 3). 4. Hypotheses of the study whether or not should be substantiated: either based on the relevant literature or on more reasons for substantiation.
--

REVIEWER	Roberts, Rossela Bangor University, Medical Sciences
REVIEW RETURNED	16-Aug-2021

GENERAL COMMENTS	Thank you for a very interesting read and a timely analysis of antipsychotic prescribing in this patient population. My comments below pertain to points of clarity, rather than methodology, and I am certain that some of these clarifications may have been left out due to word-limit constraints. For the benefit of readers not familiar with THIN, I would have liked to have seen a clarification that the data sourced from THIN pertains to approximately 6% of UK population, based on data contributed by member GP practices – in data source paragraph, page 3, line 34 onwards - and therefore, albeit the data may be representative of the UK population, it would be useful to include a statement on the generalisability of the results and acknowledgement of this limitation in the 'Strengths and weaknesses' section. Secondly, in the absence of information on how a diagnostic of PD was reached, could it be that some of these patients may have been diagnosed/treatment initiated in secondary care?
--

	Is there a duplication in introduction paragraph, page 3, between line 12 and line 16 in describing and examining time trends in prescribing antipsychotics? could the increased frequency of prescribing be an artefact of increased frequency of diagnosis in these subgroups (females are more likely to be diagnosed with EUPD and a high prevalence of adverse childhood experiences among patients diagnosed with EUPD would justify a higher comparative prevalence of prescribing) in relation to adverse childhood experiences, a paper by Williamson et al (2020) on General practice recording of adverse childhood experiences reports that only 0.4% of patients had a record of any Read code that mapped onto the ACE questionnaire, contrasting with survey- reported rates of 47% in population samples. It would be helpful to clarify Read codes that mapped in this data source, in addition to reporting on whether inferred ACEs that related to safeguarding children concerns, wider aspects of ACEs and adult consequences of ACE were added to the data capture. (covariates paragraph, page 4, line 48 onwards) In the discussion section, principal findings, page 7 row 48 onwards, in relation to the prevalence of antipsychotic prescribing, it would be useful to highlight that prescribing in PD needs to be considered on the background of an increased overall trend of antipsychotic prescribing (NCAP, 2020) To give a more rounded perspective of the problem, in the 'Interpretation, implications, conclusions and future research' section, albeit not in the direct scope of the paper's objectives, it would be useful to report on any polypharmacy of concern and whether this could be further explored, particularly in relation to outcome indicators and management of inadequate response to treatment, and the need to introduce metabolic/cardiovascular monitoring and interventions on reducing risk factors associated with the deprivation index (substance misuse)
--	--

REVIEWER	Jönsson, EG Karolinska Institutet
REVIEW RETURNED	23-Aug-2021

GENERAL COMMENTS	The authors have performed a study investigating the prescription of antipsychotics among patients with personality disorders, based on a national database on general practices in the United Kingdom. Approximately 25% of PD patients without a diagnosis of severe mental illness were prescribed antipsychotics, and the prescription rate of antipsychotics were higher if the patient lived in socially deprived areas, was a women, or had a history of adverse childhood experiences. The prescription frequency increased over time. The use of antipsychotics among patients with personality disorder is contrary to the current UK guidelines, and the authors conclude that there is need for review of the effectiveness and risks of prescribing antipsychotics to patients with personality disorders. This report has a good rationale and is of great interest. The report is relatively well-written and seems to be well conducted. See below a few suggestions. 1. Abstract: In Results section, I suggest that you describe that deprivation had to do with the socio-economic situation in the area the patient lived, only writing deprivation make it difficult to
---

	understand. You may write: “level of deprivation in the area where the patient lived” or something similar. 2. Regarding deprivation I also suggest that you use the least deprived Townsend score (1) as 1 (the “normal”), for which you compare the other Townsend scores (2 – 5) (Table 3). In the abstract you write that “prescribing increased with deprivation”, and then comes the ratio of 0.56. It would be more easily understood with a positive adjusted ratio (1.79?). 3. Introduction, page 3, 4th row. You write “austerity”. Do you mean “economic austerity”? 4. Introduction, page 3, row 6-11. You start to write that you have three aims, in practice you list four. You lack “;” between aim two and three. You write “3” both after the third and the fourth aim. Please, correct. 5. Discussion, 2nd row: Replace “are” with “were”. 6. Discussion, row 7-8: It is written: “... individuals with greater area level deprivation ...”. I suggest: “... individuals living in areas with greater deprivation ...”. 7. Page 8, first paragraph, last sentence. Is a “be” missing? 8. Page 8, 3rd paragraph, last sentence: add “years” after “18” and “11”. 9. Page 9, 2nd paragraph, last sentence: You write “austerity”. Do you mean “economic austerity”? 10. Page 9, 3rd paragraph, row 4: add “antipsychotics.” After “of”. 11. Explain abbreviations at their first appearance in the text: EUPD, UK, SMI, PPI, NHS, NIHR.
--	---

REVIEWER	Hakko, Helinä Oulu University Hospital, Department of Psychiatry
REVIEW RETURNED	19-Sep-2021

GENERAL COMMENTS	The study focus is important and up-to-date, since studies on the prevalence of PD patients in primary care settings are few. The manuscript, however, would benefit for major revision to be more clearer for international readers. The suggestions for modifications are listed below: Abstract  - Objective: The term “frequency” is not the proper expression here. INTRODUCTION  - page 1, in the sentence beginning with “Prevalence estimates for PD vary ...” (lines 33-34) report the exact prevalence rates. It helps reader to compare the prevalences of the current study to those reported in earlier study. - Spell out the abbreviation “EUPD”, because it is mentioned here for the first time (line 37) - page 2: SMI is mentioned first time here, so spell it out here (lines 11-12) - the third aim is written twice in the text - the hypothesis addressing the role of deprivation and adverse childhood experience is not needed here, because study focus is to estimate the prevalence of antipsychotic users among PD patients with and without SMI. Namely, ACEs and deprivation are covariates among all covariates used in this study. METHODS Data source:
---

Some clarification for data sources is needed for readers. First, the data comprised of 744 general practices across the UK. What is the proportion that these 744 general practices covered of all general practices in UK? Further, what means that the database is “broadly” representative of the UK general population (lines 38-39)? The THIN database needs also to be briefly described here, because readers from other countries than UK are not familiar with it.

Study population:

The major methodological concern is the use of Read codes in diagnoses instead of ICD-codes, because such coding system is not commonly used worldwide. It also remains unclear how comparable the Read codes for PD (and other mental health conditions) are with corresponding ICD-codes. Thus, for international readers, the use of ICD-10 codes would be preferable throughout the manuscript including methods, Appendix Table 1, statistical analyses etc.

The severity level of PD varies between types of PD and, consequently, need of specific types of psychotropic medication, particularly antipsychotics, varies between the types of PD, as the authors wrote in the introduction section. At current version, the results are, however, reported by specific categories of PD (a total of nine subcategories), which is rather messy. These nine PD subtypes classified to smaller groups, for example, to severity level or clusters of PD (A, B, C, nos) would give clearer and more focused view about the role of antipsychotic in treatment of PD.

Outcomes:

- Appendix Table 2 would be more reader-friendly if it clearly shows which antipsychotics belongs to first- and second-generation antipsychotics.
- The table includes all antipsychotic prescriptions and during the follow-up time a person may have received several types of antipsychotics aft.
- The type of specific PD of an individual was defined based on the most recent record, while the use of antipsychotics was explored after first PD diagnosis identified from the registry. I wonder what was the distribution of types of antipsychotics if evaluated at the time of last PD diagnosis?
- The use of 84 days as cut-off for between consecutively prescriptions indicating the longer use of antipsychotics appropriately is justified, but the literature reference is missing

Follow-up period:

- the definition for the calculation of the length of follow-up time (later referred as duration in the study) of each participant is clearly described in the text.
- Is it possible to estimate the proportion of first onset PD versus long-term PD patients?

Covariates:

- The quintiles of Townsend score is worth of briefly describing already here, because it is likely that many readers are not familiar with that scoring system.
- The use of ICD-codes instead of broad names of disorder is preferable, because it gives more accurate understanding what disorders are included in SMI. Now it is stated that SMI means that a patient has schizophrenia, bipolar disorder, or other non-organic

	psychotic illness. Did psychiatrists confirm this SMI definition? Did the authors include both Type I and Type II bipolar disorders to SMI? Usually the type II bipolar disorder is not categorized to SMI. Did the mental health condition referred as “depression” include also psychotic depression? Patient and public involvement:  - The first sentence is unclear, for example, what means that “service users were involved in the study”? Statistical analysis:  - Generally, the choice of statistical analyses is appropriate in terms of the registry data used in analyses. - The reporting style of the number of outcomes is messy, here six outcomes, see abstract and aims. - The analyses are stratified by the status of SMI (yes, no). As noted earlier, recheck which psychiatric disorders should be included in SMI and add also reference to earlier studies addressing SMI of patients. - Last sentence of the first paragraph (page 5, lines 20-21, “For the prevalence estimates of...” is unclear. Particularly, what means “... the relevant period of follow-up ...”. - Two sensitivity analyses were performed. It would be more reader-friendly that each sensitivity analysis is separately described here. - The name of the statistical software used in analyses need to be reported in the text. RESULTS  - Report the prevalence of PD patients observed in the whole data of the general practice patients. - page 5, lines 52-53, “... EUPD recorded among 12 300 (27% of the cohort).” The term “cohort” may be confusing here because it can also be understand to mean all persons recorded in the THIN, not just those with PD. Thus, use the term “PD cohort” always when referring to PD patients. - In addition to age at study entry, the age at first as well as last PD diagnosis recorded is worth of reporting in the table 1. - the average age at entry was 43 years. Because all types of PDs were summed up, it is likely that average age at entry varied between the severity of PD. The average ages by clusters of PD would be useful information for readers when interpreting the results. Prevalence of antipsychotic prescribing :  - The actual prevalence rates of the use of first- and second-generation antipsychotics, separately, would be important information in the result section. - Further, did the results of table 3 remained the same, if the analyses were stratified by the use of first and second-generation antipsychotics? Time trends:  - Does “The frequency of prescribing ...” refer to number of all antipsychotic prescriptions? If so, this figure might be misleading because a patient may have received only one antipsychotic prescription versus several prescriptions during follow-up time? - The type of specific PD was defined from the last PD diagnosis entry to the data. If the type of antipsychotic has been evaluated
--	--

	based on the last PD diagnosis entry, I wonder whether this time trend figure has changed!? Individual antipsychotics agents: - Appendix table 2 refer the total number of antipsychotic prescriptions, which assumable is higher than the number of individual PD patients with antipsychotics. If so, or otherwise, this need to be stated clearly throughout the manuscript. DISCUSSION: Whether the discussion is supported by the data remains unclear, because of abovementioned methodological concerns, the major ones being to the use of Read codes for PD instead of ICD-codes for PD and definition for SMI.
--	---

REVIEWER	Boland, Fiona Royal College of Surgeons Ireland, 123 St Stephens Green, HRB Centre For Primary Care Research, Division of Population Health Sciences (PHS)
REVIEW RETURNED	27-Sep-2021

GENERAL COMMENTS	Overall I found the manuscript to be an interesting read. It is well written and has a good structure. Study aims: the aim around time trends has been included twice as aim 2) and 3). In relation to the follow-up period, could you elaborate on 'date practice reached data quality thresholds'? Did the authors assess this? Were the dates derived by Maguire et al and Horsfall et al used? In the abstract and other areas, the study end date is Dec 2017. However, in others places, for example the follow-up period and time trends, it states Dec 2016. Please clarify. It is not entirely clear in the text whether the primary outcome measure is a count variable or binary? In the outcome section, and Table 3, it would suggest it is a binary variable, that is, at least one prescription for antipsychotic medication. Is this calculated at the individual level and over the entire study period? Was logistic regression considered? Was time in the study adjusted / accounted for in the models? Was there a minimum follow-up time required for individuals? Did you explore significant differences over time in the prevalence of antipsychotic prescribing? Possibly a negative binomial regression could be used to quantify the change in the rate associated with study year, controlling for other variables. In the main results section, it would help to elaborate on the results of the models – specifically Table 3 and the interpretation of the rate ratios.
--

VERSION 1 – AUTHOR RESPONSE

Reviewer: 1

Dr. Alexander M. Ponizovsky, Israeli Ministry of Health

Comments to the Author:

Minor points

1. All abbreviations should be fully explained in the first mention in the text.

All abbreviations checked and amended where needed (EUPD, UK, SMI, PPI, NHS, NIHR)

2. Abstract. Investigate should be read To investigate...

Done (Page 1)

3. Introduction. Aims 2 is duplicated (see aim 3).

Thank you for alerting us to this. Done – first mention removed so aims given in same order as results (Page 3)

4. Hypotheses of the study whether or not should be substantiated: either based on the relevant literature or on more reasons for substantiation.

We have removed the hypothesis in line with comments from Reviewer 4.

Reviewer: 2

Dr. Rossela Roberts, Bangor University, Health and Care Research Wales

Comments to the Author:

Thank you for a very interesting read and a timely analysis of antipsychotic prescribing in this patient population.

My comments below pertain to points of clarity, rather than methodology, and I am certain that some of these clarifications may have been left out due to word-limit constraints.

For the benefit of readers not familiar with THIN, I would have liked to have seen a clarification that the data sourced from THIN pertains to approximately 6% of UK population, based on data contributed by member GP practices – in data source paragraph, page 3, line 34 onwards ; The first paragraph of the Data Source section has been amended as follows (Page 3):

We used data from The Health Improvement Network (THIN) primary care database, which comprised, at time of analysis, computerised anonymised patient records retrieved from 744 member general practices across the UK, corresponding to approximately 6% of the UK population.

and therefore, albeit the data may be representative of the UK population, it would be useful to include a statement on the generalisability of the results and acknowledgement of this limitation in the 'Strengths and weaknesses' section.

The beginning of the Strengths and Weaknesses section has been amended as follows (Page 9):

The strengths of our study include the large size of the cohort, enabling precise estimates, and UK-wide coverage. While the THIN database pertains to approximately 6% of the UK population, previous studies have shown THIN to be socially and demographically representative of wider UK general practice (Blak et al), thus, the results may be cautiously generalised beyond the THIN population to the wider UK general practice population.

Secondly, in the absence of information on how a diagnosis of PD was reached, could it be that some of these patients may have been diagnosed/treatment initiated in secondary care?

We agree it is possible that some patients may have been diagnosed and treatment initiated in secondary care. Where a person received a diagnosis in secondary care, but the diagnosis was not captured in the primary care records, they would not have been included in the study sample. Where treatment was initiated in secondary care, either treatment will have continued in primary care, and the person would have been included in the treatment counts, or the treatment did not continue in primary care, in which case the effect would have been to underestimate the extent of antipsychotic use in the study sample. However focus in this study is on prescribing practices specifically in the primary care setting so in this sense diagnosis/ treatment in secondary care does not affect the validity of the results, but may offer an explanation for the findings, as noted in the Discussion (Page 10):

It may be that the prescriptions issued in primary care are frequently provided as follow-on prescribing, after an initial prescription by a secondary care psychiatrist, without clear guidance on when to stop prescribing the antipsychotic drugs.

Is there a duplication in introduction paragraph, page 3, between line 12 and line 16 in describing and examining time trends in prescribing antipsychotics?

Thank you for alerting us to this. Done – first mention removed so aims given in same order as results (Page 3)

could the increased frequency of prescribing be an artefact of increased frequency of diagnosis in these subgroups (females are more likely to be diagnosed with EUPD and a high prevalence of adverse childhood experiences among patients diagnosed with EUPD would justify a higher comparative prevalence of prescribing)

We agree that increased diagnosis of EUPD, or increased reporting of adverse childhood experiences, may help to explain the increase in prescribing over time and have added this to the comment on the time trends in the Discussion (Page 10):

The increase in antipsychotic prescriptions over time could reflect the impact of longer waiting lists for psychological interventions during a period of austerity, as well as general trends in prescribing of antipsychotics that has been observed for other conditions, and increased diagnosis of EUPD and increased reporting of adverse childhood experiences, which were associated with higher levels of prescribing.

While increased diagnosis of EUPD, or increased reporting of adverse childhood experiences, may help to explain the increase in prescribing over time, they do not justify the increase in prescribing. This is because current guidelines do not recommend use of antipsychotics in PD in general, regardless of/ including for specific diagnoses or history of adverse childhood experiences. Indeed, as noted in the Discussion (Page 10), there is some evidence to suggest those with adverse childhood experiences, may benefit more from appropriate psychological interventions (Nemeroff et al).

in relation to adverse childhood experiences, a paper by Williamson et al (2020) on General practice recording of adverse childhood experiences reports that only 0.4% of patients had a record of any Read code that mapped onto the ACE questionnaire, contrasting with survey- reported rates of 47% in population samples. It would be helpful to clarify Read codes that mapped in this data source, in addition to reporting on whether inferred ACEs that related to safeguarding children concerns, wider aspects of ACEs and adult

consequences of ACE were added to the data capture. (covariates paragraph, page 4, line 48 onwards)

The ACE code list included codes corresponding to items in the Centers for Disease Control and Prevention short ACEs tool and codes corresponding to safeguarding of children/ contact with social services. The code list was based on previous work examining how ACEs are recorded in THIN (Woodman et al). We have rewritten this section to give more details of the types of records included and how the code list was developed (Pages 4-5):

The presence of recorded adverse childhood experiences was identified from records with Read codes specifically relating to: contact with social services, child protection procedures, child on at risk register, child-at-risk case conference, out-of-home care, or codes corresponding to items in the Centers for Disease Control and Prevention short adverse childhood experiences tool (Anda et al) that is; direct reference to childhood maltreatment (physical or emotional abuse or neglect), or direct reference to household dysfunction (such as parental substance misuse). Codes included were either i) codes recorded in childhood (when individual <17 years) or ii) codes recorded during adulthood indicative of a history of the above in childhood. The code list was based on previous work examining recording of childhood maltreatment in THIN(19).

Given the difficulty and uncertainty in establishing which codes refer to ACE, we therefore also carried out sensitivity analyses using different definitions: i) limited to records referring specifically to contact with social services in childhood and ii) limited to individuals present in the database from childhood and therefore excluding retrospective reporting of historical ACE (Pages 7-8).

In the discussion section, principal findings, page 7 row 48 onwards, in relation to the prevalence of antipsychotic prescribing, it would be useful to highlight that prescribing in PD needs to be considered on the background of an increased overall trend of antipsychotic prescribing (NCAP, 2020) We note in the Discussion (Page 10):

The increase in antipsychotic prescriptions over time could reflect the impact of longer waiting lists for psychological interventions during a period of austerity, as well as general trends in prescribing of antipsychotics that has been observed for other conditions

To give a more rounded perspective of the problem, in the 'Interpretation, implications, conclusions and future research' section, albeit not in the direct scope of the paper's objectives, it would be useful to report on any polypharmacy of concern and whether this could be further explored, particularly in relation to outcome indicators and management of inadequate response to treatment, and the need to introduce metabolic/cardiovascular monitoring and interventions on reducing risk factors associated with the deprivation index (substance misuse)

We agree polypharmacy could pose an additional issue associated with unmonitored prescribing of antipsychotics in this population and have added this concern to the Discussion (Page 9):

The people in our cohort receiving antipsychotics will be at risk of experiencing adverse side-effects from the drugs, possibly without any proven long term benefit of these agents for their mental health. They may develop adverse physical effects and yet may not receive appropriate physical health monitoring for weight gain and metabolic disturbance, because in the UK NHS physical health monitoring is currently only targeted at people with SMI. The lack of formal monitoring and protocols for prescribing antipsychotics in PD could also put people at risk of adverse effects of polypharmacy (for example other psychotropic medications or medications for physical long term conditions).

Reviewer: 3

Dr. EG Jönsson, Karolinska Institutet

Comments to the Author:

The authors have performed a study investigating the prescription of antipsychotics among patients with personality disorders, based on a national database on general practices in the United Kingdom. Approximately 25% of PD patients without a diagnosis of severe mental illness were prescribed antipsychotics, and the prescription rate of antipsychotics were higher if the patient lived in socially deprived areas, was a women, or had a history of adverse childhood experiences. The prescription frequency increased over time. The use of antipsychotics among patients with personality disorder is contrary to the current UK guidelines, and the authors conclude that there is need for review of the effectiveness and risks of prescribing antipsychotics to patients with personality disorders.

This report has a good rationale and is of great interest. The report is relatively well-written and seems to be well conducted. See below a few suggestions.

1. Abstract: In Results section, I suggest that you describe that deprivation had to do with the socio-economic situation in the area the patient lived, only writing deprivation make it difficult to understand. You may write: "level of deprivation in the area where the patient lived" or something similar.

The Abstract has been amended to (Page 1):

Prescribing was lower in less deprived areas

2. Regarding deprivation I also suggest that you use the least deprived Townsend score (1) as 1 (the "normal"), for which you compare the other Townsend scores (2 – 5) (Table 3). In the abstract you write that "prescribing increased with deprivation", and then comes the ratio of 0.56. It would be more easily understood with a positive adjusted ratio (1.79?).

The most deprived quintile was used as the baseline category to ensure stability of risk ratio estimates as it is the largest category, while the least deprived quintile is the smallest (Table 1). We agree that the ratio 0.56 is somewhat counterintuitive when discussing the elevated risk in more deprived areas and so have altered the Abstract to read (Page 1):

Prescribing was lower in less deprived areas

3. Introduction, page 3, 4th row. You write "austerity". Do you mean "economic austerity"?
Yes – we have added "economic" before each mention of austerity for clarity (Page 3 and Page 10)

4. Introduction, page 3, row 6-11. You start to write that you have three aims, in practice you list four. You lack ";" between aim two and three. You write "3" both after the third and the fourth aim.
Please, correct.

We have removed the repeated aim on time trends and corrected the numbering

5. Discussion, 2nd row: Replace "are" with "were".
Done

6. Discussion, row 7-8: It is written: "... individuals with greater area level deprivation ...". I suggest: "... individuals living in areas with greater deprivation ...".
Done

7. Page 8, first paragraph, last sentence. Is a "be" missing?
Yes – corrected

8. Page 8, 3rd paragraph, last sentence: add "years" after "18" and "11".

Done

9. Page 9, 2nd paragraph, last sentence: You write “austerity”. Do you mean “economic austerity”?
Yes – corrected to read “economic austerity”

10. Page 9, 3rd paragraph, row 4: add “antipsychotics.” After “of”.
Done

11. Explain abbreviations at their first appearance in the text: EUPD, UK, SMI, PPI, NHS, NIHR.
Done

Reviewer: 4
Dr. Helinä Hakko, Oulu University Hospital

Comments to the Author:

The study focus is important and up-to-date, since studies on the prevalence of PD patients in primary care settings are few. The manuscript, however, would benefit for major revision to be more clearer for international readers. The suggestions for modifications are listed below:

We thank the Reviewer for their insightful, detailed comments.

Abstract

- Objective: The term “frequency” is not the proper expression here.
The term “frequency” has been replaced with “extent”

INTRODUCTION

- page 1, in the sentence beginning with “Prevalence estimates for PD vary” (lines 33-34) report the exact prevalence rates. It helps reader to compare the prevalences of the current study to those reported in earlier study.

This sentence has been expanded as follows (Page 2):

Prevalence of PD in the community is estimated to be 6-10% but estimates vary globally and by clinical setting (2)

- Spell out the abbreviation “EUPD”, because it is mentioned here for the first time (line 37)
Done

- page 2: SMI is mentioned first time here, so spell it out here (lines 11-12)
Done

- the third aim is written twice in the text

Thank you for alerting us to this. Done – first mention removed so aims given in same order as results (Page 3)

- the hypothesis addressing the role of deprivation and adverse childhood experience is not needed here, because study focus is to estimate the prevalence of antipsychotic users among PD patients with and without SMI. Namely, ACEs and deprivation are covariates among all covariates used in this study.

We agree and have removed the hypothesis.

METHODS

Data source:

Some clarification for data sources is needed for readers. First, the data comprised of 744 general practices across the UK. What is the proportion that these 744 general practices covered of all general practices in UK? Further, what means that the database is “broadly” representative of the UK general population (lines 38-39)? The THIN database needs also to be briefly described here, because readers from other countries than UK are not familiar with it.

The Data Source section has been rewritten to address these points as follows (Page 3):

We used data from The Health Improvement Network (THIN) primary care database, which is a clinical database comprising, at time of analysis, computerised anonymised patient records retrieved from 744 member general practices across the UK, corresponding to approximately 6% of the UK population. Patient records are longitudinal, including all information recorded by the GP from when the patient registered with the member practice until the date the practice last contributed data to THIN or the patient left the practice. Records for a patient are linked via an anonymised patient identifier.

The database includes all diagnostic, symptoms and other health data (blood tests, health indicators) recorded by the GP in the patient’s computerised records, as well as all prescriptions issued. Diagnoses, symptoms and other health information in THIN are coded using the Read code clinical classification system. The database has been shown to be socially and demographically representative of the UK general practice population.

Study population:

The major methodological concern is the use of Read codes in diagnoses instead of ICD-codes, because such coding system is not commonly used worldwide. It also remains unclear how comparable the Read codes for PD (and other mental health conditions) are with corresponding ICD-codes. Thus, for international readers, the use of ICD-10 codes would be preferable throughout the manuscript including methods, Appendix Table 1, statistical analyses etc.

We recognise that this is a limitation of the THIN database. However, the Read codes for PD were identified by cross-referencing with ICD-10 chapters F60-F61. This is noted in the Study population section (Pages 3 -4). Further, the complete list of Read codes used to identify PD are given in the Appendix so that readers unfamiliar with the Read code system are able to ascertain how the study sample was defined. As the Read codes map onto ICD-10 codes, for clarity the labelling of “diagnosis category of PD” in the tables has been altered to: “ICD-10 diagnostic category of PD”.

The severity level of PD varies between types of PD and, consequently, need of specific types of psychotropic medication, particularly antipsychotics, varies between the types of PD, as the authors wrote in the introduction section. At current version, the results are, however, reported by specific categories of PD (a total of nine subcategories), which is rather messy. These nine PD subtypes classified to smaller groups, for example, to severity level or clusters of PD (A, B, C, nos) would give clearer and more focused view about the role of antipsychotic in treatment of PD.

The nine PD categories were pre-specified prior to the analysis, based on ICD-10 categories, and kept separate as they generally correspond to quite distinct patient profiles. We did not want to combine categories to avoid potentially masking differences in prescribing between categories. There are not codes for clusters in the UK or markers for severity. We have added as a limitation that the clusters used in other international settings are not coded in UK primary care (Page 9):

A further limitation is that the clusters of PD severity used in other international settings are not coded in UK primary care.

In future work we would like to examine prescribing by PD category in more detail, including according to severity levels or clusters. This has been added to the section on future research (Page 10):

Investigations of antipsychotic prescribing according to PD severity clusters is also warranted.

Outcomes:

- Appendix Table 2 would be more reader-friendly if it clearly shows which antipsychotics belongs to first- and second-generation antipsychotics.

Agree – this has been amended

- The table includes all antipsychotic prescriptions and during the follow-up time a person may have received several types of antipsychotics aft.

This is correct and a footnote to the table indicates that this is the case to alert the reader.

- The type of specific PD of an individual was defined based on the most recent record, while the use of antipsychotics was explored after first PD diagnosis identified from the registry. I wonder what was the distribution of types of antipsychotics if evaluated at the time of last PD diagnosis?

This is a very interesting point. However, the majority of people in the study only had one type of PD diagnosis so it is not possible to check this. Only a small number of people had more than one PD diagnosis type and then the most recent record was used to determine the type of PD, on the assumption that the most recent diagnosis was the more reliable. Therefore, given the small numbers affected, the results would be little altered.

- The use of 84 days as cut-off for between consecutively prescriptions indicating the longer use of antipsychotics appropriately is justified, but the literature reference is missing

The use of 84 days was a pragmatic choice, reflecting standard prescription issue length in UK primary care i.e. 28 days x 3 prescriptions.

Follow-up period:

- the definition for the calculation of the length of follow-up time (later referred as duration in the study) of each participant is clearly described in the text.

- Is it possible to estimate the proportion of first onset PD versus long-term PD patients? A limitation of THIN is that it only contains information for the period when a patient is registered with a member practice., and patients may move in and out of these practices. Therefore it is challenging to be certain that a patient had first onset or long term PD. Thus the data tell us which people with diagnosed PD receive antipsychotic medications but not how far after their initial diagnosis they receive these medications. This would make for interesting future analyses.

Covariates:

- The quintiles of Townsend score is worth of briefly describing already here, because it is likely that many readers are not familiar with that scoring system.

Done – The first sentence in the Covariates section has been expanded as follows (Page 4):

Antipsychotic prescribing was explored according to the following sociodemographic characteristics: gender, age and Townsend score for area-level social deprivation for the area (postcode) in which the individual lived (grouped into quintiles)(Townsend). The Townsend score incorporates four variables: percentage of adults unemployed, percentage non-car ownership, percentage non-home ownership and percentage overcrowded households.

- The use of ICD-codes instead of broad names of disorder is preferable, because it gives more accurate understanding what disorders are included in SMI. Now it is stated that SMI means that a patient has schizophrenia, bipolar disorder, or other non-organic psychotic illness. Did psychiatrists confirm this SMI definition? Did the authors include both Type I and Type II bipolar disorders to

SMI? Usually the type II bipolar disorder is not categorized to SMI. Did the mental health condition referred as “depression” include also psychotic depression?

The Read code list used to identify SMI is (an update of) an established list that has been used and validated previously for studies on SMI in THIN by the authors: (Osborn et al. JAMA Psychiatry. 2015 Feb;72(2):143-51., Hardoon et al. PLoS One. 2013 Dec 12;8(12):e82365).

Two authors (JFH and DPJO) are psychiatrists who were involved in developing this SMI definition. Bipolar II was included in SMI in this instance as it is an acceptable indication for antipsychotics and the purpose was to identify anyone who may have a mental health diagnosis, in addition to their PD, for which antipsychotics could be indicated. Thus we could be as certain as possible that the group with no SMI had no indication for antipsychotic use, supporting the key message on prescribing in PD counter to current guidelines. Psychotic depression was also included as SMI.

Patient and public involvement:

- The first sentence is unclear, for example, what means that “service users were involved in the study”?

We have rephrased this section to make it clearer (Page 5):

A group of service users was established in conjunction with the McPin Foundation, called the Pathfinder Data Science PPI Group. Through regular meetings over the course of the study, members of the group were involved in the design, conduct and reporting plans for our research.

Statistical analysis:

- Generally, the choice of statistical analyses is appropriate in terms of the registry data used in analyses.

- The reporting style of the number of outcomes is messy, here six outcomes, see abstract and aims.

- The analyses are stratified by the status of SMI (yes, no). As noted earlier, recheck which psychiatric disorders should be included in SMI and add also reference to earlier studies addressing SMI of patients.

Please see above – reference to previous study now included (Page 4) (Hardoon et al. PLoS One. 2013 Dec 12;8(12):e82365).

- Last sentence of the first paragraph (page 5, lines 20-21, “For the prevalence estimates of...” is unclear. Particularly, what means “... the relevant period of follow-up ...”.

We have amended the first paragraph of the Statistical Analysis to make it clearer:

Statistical Analysis section (Page 5):

We first determined the percentages of people: i) receiving at least one antipsychotic prescription, ii) at least two antipsychotic prescriptions fewer than 84 days apart. We also determined percentages of people with a continuous treatment period of at least i) six months, ii) one year, iii) three years and iv) five years. Percentages were computed....For the continuous treatment periods percentages calculations, the study sample was restricted to only individuals who had at least that length of follow-up in the database (six months, one year, three years or five years).

- Two sensitivity analyses were performed. It would be more reader-friendly that each sensitivity analysis is separately described here.

We have amended this section to describe the sensitivity analyses separately and more clearly (Page 5):

We carried out two sensitivity analyses using different definitions of adverse childhood experiences. In the first sensitivity analysis, we restricted the definition of adverse childhood experiences to a record (current or retrospective) of contact with social services before age 17

years. In the second sensitivity analysis, we only included people with primary care data available from childhood (from at least age 11 years), to enable capture of adverse childhood experiences as recorded during childhood, rather than historically.

- The name of the statistical software used in analyses need to be reported in the text.
Done – Final sentence added to Statistical Analysis section (Page 5):

Stata (version 14) was used for all analyses.

RESULTS

- Report the prevalence of PD patients observed in the whole data of the general practice patients. We only extracted records for PD patients, not all patients in the database, so do not have denominators to ascertain prevalence, as this is not one of the study aims.

- page 5, lines 52-53, "... EUPD recorded among 12 300 (27% of the cohort)." The term "cohort" may be confusing here because it can also be understand to mean all persons recorded in the THIN, not just those with PD. Thus, use the term "PD cohort" always when referring to PD patients.
Agree – have replaced "cohort" with "PD cohort" throughout, including in the Methods, when describing the study population.

- In addition to age at study entry, the age at first as well as last PD diagnosis recorded is worth of reporting in the table 1.
Given the afore-mentioned difficulty in establishing whether the first PD record is truly the first diagnosis of PD, we limit reporting of age to study entry as we cannot confidently state the date of first PD diagnosis.

- the average age at entry was 43 years. Because all types of PDs were summed up, it is likely that average age at entry varied between the severity of PD. The average ages by clusters of PD would be useful information for readers when interpreting the results. As above, a more detailed examination of clusters of severity is beyond the scope of this paper, but we agree that this would be important to explore in future analyses, to elucidate reasons for the antipsychotic prescribing and have added this to the section on future research (Page 10):

Investigations of antipsychotic prescribing according to PD severity clusters is also warranted.

Prevalence of antipsychotic prescribing :

- The actual prevalence rates of the use of first- and second-generation antipsychotics, separately, would be important information in the result section.

We haven't broken down the primary results into first and second generation as our primary aim was to answer the question: are people with PD receiving any antipsychotics of any type? This is because no antipsychotics, regardless of whether they are first or second generation, are recommended in PD.

Also, analyses (and interpretation) by type of antipsychotic are complicated by the fact that people can receive more than one type. Numbers of the different types of antipsychotics prescribed to the PD cohort are instead presented in an Appendix table.

- Further, did the results of table 3 remained the same, if the analyses were stratified by the use of first and second-generation antipsychotics?

As above, we haven't broken down the primary results by type of antipsychotic. However, since second generation antipsychotics are far more commonly prescribed in the cohort (Appendix table), we would expect the patterns in table 3 to largely reflect patterns for second generation antipsychotics. We have added the following in the Strengths and Weaknesses section (Page 9):

Results were not broken down by type of antipsychotic (first or second generation). However, since second generation antipsychotics are far more commonly prescribed in the cohort (Appendix table), we would expect the patterns in prescribing of all antipsychotics to largely reflect patterns for second generation antipsychotics

Time trends:

- Does "The frequency of prescribing ..." refer to number of all antipsychotic prescriptions? If so, this figure might be misleading because a patient may have received only one antipsychotic prescription versus several prescriptions during follow-up time?

No, the "frequency of prescribing" does not refer the number of all antipsychotic prescriptions, as we agree that this would be misleading. Instead, the "frequency of prescribing" refers to the percentage of the PD cohort prescribed at least one antipsychotic in each calendar year, computed as: $100 \times (\text{number of people from PD cohort present in that calendar year with at least one antipsychotic prescription in that calendar year}) / (\text{total number of people from cohort present in that calendar year})$. This is detailed in the Statistical Analysis section and in the Figure as a footnote. However we agree it is not clear in the Results Section on Time Trends and have now amended this section as follows (Page 7):

Figure 1 presents for each calendar year, the percentage of people in the PD cohort prescribed at least one antipsychotic in that year. The percentage of people receiving prescriptions increased over time: Of those individuals who were present in the PD cohort in 2000, 17% received antipsychotics in that year...

- The type of specific PD was defined from the last PD diagnosis entry to the data. If the type of antipsychotic has been evaluated based on the last PD diagnosis entry, I wonder whether this time trend figure has changed!?

This is an interesting question, thank you. As above however, given the small numbers of people who would have more than one type of PD diagnosis, this is difficult to explore, and would have minimal impact on the results.

Individual antipsychotics agents:

- Appendix table 2 refer the total number of antipsychotic prescriptions, which assumable is higher than the number of individual PD patients with antipsychotics. If so, or otherwise, this need to be stated clearly throughout the manuscript.

Yes this is the case. This is outlined in a footnote in the Appendix Table, but we agree it should be highlighted in the main document. The following sentence has been added to the Individual antipsychotic agents section to this effect (Page 7):

Individuals may receive more than one type of antipsychotic agent so the percentages in Appendix table 2 add up to more than 100%.

DISCUSSION:

Whether the discussion is supported by the data remains unclear, because of abovementioned methodological concerns, the major ones being to the use of Read codes for PD instead of ICD-codes for PD and definition for SMI.

We have detailed above the rationale for our definitions of PD and SMI in the study. In particular, we believe that it is important to use the broadest definition of SMI to rule out any possible indication for antipsychotic prescribing in the no SMI group. Furthermore, while we appreciate that in a clinical effectiveness study, precise reproducible/ universal definitions are crucial, in this observational study designed to explore GP prescribing practices, we are addressing the question: if a GP has indicated

that their patient has PD through the patient receiving an ICD-10 related primary care Read code for PD in their primary care records, does the patient also receive antipsychotic medications from the GP? So most important is that our definition of PD reflects the GP definition.

Reviewer: 5

Dr. Fiona Boland, Royal College of Surgeons Ireland, 123 St Stephens Green

Comments to the Author:

Overall I found the manuscript to be an interesting read. It is well written and has a good structure.

Study aims: the aim around time trends has been included twice as aim 2) and 3).

Thank you for alerting us to this. Done – first mention removed so aims given in same order as results (Page 3).

In relation to the follow-up period, could you elaborate on 'date practice reached data quality thresholds'? Did the authors assess this? Were the dates derived by Maguire et al and Horsfall et al used?

Yes, these were the established dates derived by Maguire et al and Horsfall et al. We have amended the Study Population Section (Page 3):

All permanently registered patients were included from the date the general practice reached previously defined, established data quality thresholds: acceptable mortality rate date (Maguire et al) and acceptable computer usage date (Horsfall et al)

And have amended the Follow-up Period section (Page 4):

Cohort entry was defined as the latest of: registration at the general practice, the established practice acceptable mortality rate (Maguire et al), the established practice acceptable computer usage date (Horsfall et al), date of first PD diagnosis record, or 1 January 2000

In the abstract and other areas, the study end date is Dec 2017. However, in others places, for example the follow-up period and time trends, it states Dec 2016. Please clarify.

The study end date is Dec 2016. This has been corrected throughout.

It is not entirely clear in the text whether the primary outcome measure is a count variable or binary? In the outcome section, and Table 3, it would suggest it is a binary variable, that is, at least one prescription for antipsychotic medication. Is this calculated at the individual level and over the entire study period? Was logistic regression considered? Was time in the study adjusted / accounted for in the models? Was there a minimum follow-up time required for individuals?

The primary outcome (in table 3 and third paragraph of Prevalence of antipsychotic prescribing section) is the rate of antipsychotic medication prescribing, computed as: total people receiving a prescription/ total follow-up time for people in each covariate category. In this way follow-up time is accounted for in the modelling. This outcome was modelled using Poisson regression, which takes into account follow-up time for each individual. It is an individual-level analysis using the total follow-up time for each individual as an offset.

In addition, table 2 and the first two paragraphs of Prevalence of antipsychotic prescribing section present percentages of people in receipt of antipsychotic medication, ignoring follow-up time, as additional descriptive statistics.

We agree that this distinction has not been expressed clearly in the manuscript and have amended the Statistical Analysis section and Prevalence of antipsychotic prescribing section to make this clearer:

Statistical Analysis section (Page 5):

We first determined the percentages of people: i) receiving at least one antipsychotic prescription, ii) at least two antipsychotic prescriptions fewer than 84 days apart. We also determined percentages of people with a continuous treatment period of at least i) six months, ii) one year, iii) three years and iv) five years. Percentages were computed.... For the continuous treatment periods percentages calculations, the study sample was restricted to only individuals who had at least that length of follow-up in the database (six months, one year, three years or five years).

We then used univariable and multivariable Poisson regression, with follow-up time for individuals as an offset, to estimate rate ratios for antipsychotic prescribing, to take account of differing follow-up times for individuals in the study. Rates were computed as total people receiving a prescription/ total follow-up time for people in each covariate category. Regressions were stratified by....

Prevalence of antipsychotic prescribing section (Page 6)

Percentages of people in the PD cohort prescribed antipsychotics by age, gender...are shown in Table 2.

Did you explore significant differences over time in the prevalence of antipsychotic prescribing? Possibly a negative binomial regression could be used to quantify the change in the rate associated with study year, controlling for other variables.

As the time-trend analysis was not the primary aim of the study we did not explore whether the trends over time were statistically significant to avoid carrying out too many significance tests in the one study. However, although beyond the scope of this study, we agree that a more detailed exploration of the time trends is warranted and would make for interesting future analyses.

In the main results section, it would help to elaborate on the results of the models – specifically Table 3 and the interpretation of the rate ratios.

We have expanded the third paragraph in the Prevalence of antipsychotic prescribing section (Page 6) as follows:

Adjusted and unadjusted ratios comparing rates of prescribing of antipsychotics for different levels of socio-demographic characteristics, are shown in Table 3. In the group with no additional SMI, the differences in prescribing of antipsychotics by gender, age, deprivation, type of PD and record of adverse childhood experiences all remained significant after adjustment: The adjusted rate ratio (aRR) comparing women to men was 1.25 (95% CI 1.16 to 1.34, $p < 0.001$). The aRR comparing those over 75 years to those aged 35-44 years was 2.02 (1.75 to 2.34, $p < 0.001$). The aRR comparing the least to most deprived areas was 0.56 (0.48 to 0.66, $p < 0.001$). Rates for all diagnostic categories of PD were statistically significantly lower than for EUPD with aRRs relative to EUPD of between 0.19 (0.15 to 0.23, $p < 0.001$) for anankastic PD and 0.63 (0.53 to 0.74, $p < 0.001$) for paranoid PD. The aRR comparing history to no history of adverse childhood experiences was 1.41 (1.28 to 1.56, $p < 0.001$).

VERSION 2 – REVIEW

REVIEWER	Roberts, Rossela Bangor University, Medical Sciences
REVIEW RETURNED	06-Dec-2021
GENERAL COMMENTS	Thank you for your detailed response, this has now addressed all my initial comment
REVIEWER	Hakko, Helinä Oulu University Hospital, Department of Psychiatry
REVIEW RETURNED	25-Nov-2021
GENERAL COMMENTS	The authors have satisfactory revised their manuscript according to the comments of the reviewers.
REVIEWER	Boland, Fiona Royal College of Surgeons Ireland, 123 St Stephens Green, HRB Centre For Primary Care Research, Division of Population Health Sciences (PHS)
REVIEW RETURNED	08-Dec-2021
GENERAL COMMENTS	The authors appear to have addressed all comments in the revised version. One additional comment, were specific guidelines followed in the reporting of this study (i.e. STROBE guidelines)? I don't see any mention of it in the text.